# Zebrafish Feed Intake: A Systematic Review for Standardizing Feeding Management in Laboratory Conditions

**DOI:** 10.3390/biology13040209

**Published:** 2024-03-23

**Authors:** Rosario Licitra, Baldassare Fronte, Tiziano Verri, Maria Marchese, Chiara Sangiacomo, Filippo Maria Santorelli

**Affiliations:** 1Department of Neurobiology and Molecular Medicine, IRCCS Stella Maris Foundation, 56128 Pisa, Italy; maria.marchese@fsm.unipi.it; 2Department of Veterinary Sciences, University of Pisa, 56124 Pisa, Italy; baldassare.fronte@unipi.it (B.F.); chiara.sangiacomo@phd.unipi.it (C.S.); 3Department of Biological and Environmental Sciences and Technologies, University of Salento, 73100 Lecce, Italy; tiziano.verri@unisalento.it

**Keywords:** zebrafish nutrition, feed intake, feed ingestion, feeding protocol, feeds, fish diets

## Abstract

**Simple Summary:**

Zebrafish have emerged as invaluable model organisms in biological research, offering a cost-effective alternative to rodents. However, the absence of standardized feeding protocols and nutritional requirements poses a significant challenge, potentially compromising the validity of experimental outcomes, particularly in preclinical studies. Our systematic review analyzes a plethora of studies focused on zebrafish feed intake, feeding regimes, and diet compositions. We uncover substantial variability in dietary parameters, notably in crude protein and lipid content. Despite this diversity, energy levels remain quite consistent across zebrafish diets. By using these insights, we propose a novel feeding protocol for zebrafish of different developmental stages, based on a formulated feed optimized for protein and energy content. This protocol might represent the first step towards standardizing zebrafish feeding practices, thereby enhancing experimental reproducibility and data reliability.

**Abstract:**

Zebrafish are one of the most used animal models in biological research and a cost-effective alternative to rodents. Despite this, nutritional requirements and standardized feeding protocols have not yet been established for this species. This is important to avoid nutritional effects on experimental outcomes, and especially when zebrafish models are used in preclinical studies, as many diseases have nutritional confounding factors. A key aspect of zebrafish nutrition is related to feed intake, the amount of feed ingested by each fish daily. With the goal of standardizing feeding protocols among the zebrafish community, this paper systematically reviews the available data from 73 studies on zebrafish feed intake, feeding regimes (levels), and diet composition. Great variability was observed regarding diet composition, especially regarding crude protein (mean 44.98 ± 9.87%) and lipid content (9.91 ± 5.40%). Interestingly, the gross energy levels of the zebrafish diets were similar across the reviewed studies (20.39 ± 2.10 kilojoules/g of feed). In most of the reviewed papers, fish received a predetermined quantity of feed (feed supplied). The authors fed the fish according to the voluntary intake and then calculated feed intake (FI) in only 17 papers. From a quantitative point of view, FI was higher than when a fixed quantity (pre-defined) of feed was supplied. Also, the literature showed that many biotic and abiotic factors may affect zebrafish FI. Finally, based on the FI data gathered from the literature, a new feeding protocol is proposed. In summary, a daily feeding rate of 9–10% of body weight is proposed for larvae, whereas these values are equal to 6–8% for juveniles and 5% for adults when a dry feed with a proper protein and energy content is used.

## 1. Introduction

The zebrafish (*Danio rerio*) is a freshwater teleost endemic to the Himalayan region. It lives in a broad range of water habitats, from slow-moving streams to stagnant rice paddies [1]. Scientific interest in zebrafish began in the 1960s at the University of Oregon, where the molecular biologist George Streisinger studied their nervous system. An article from Streisinger and colleagues published in the prestigious journal *Nature* in 1981 brought the zebrafish to worldwide attention, as he was able to produce clones of homozygous diploid zebrafish [2]. Streisinger was the first scientist in the world to clone a vertebrate, becoming the “Founding Father” of zebrafish research. Nowadays, the scientific value of zebrafish has been demonstrated over a broad range of biological research areas, and it has become one of the world’s most used animal models, being a cost-effective and efficient alternative to rodents [3,4]. Among others, key factors have also been its small size (3–5 cm in adulthood), egg transparency and external embryo development, ability to regenerate complex tissues, simple management, high fecundity, fast growth, and short life cycle [5,6,7]. The zebrafish life cycle consists of four major developmental stages: embryo, larva, juvenile, and adult. The cycle begins when parents release eggs and sperm. Fertilized eggs hatch after 48–96 h post-fertilization, generating free-swimming larvae. Sexual maturity is reached in approximately 90 days [8,9]. Zebrafish lifespan is 3.5 years on average, but in some cases, they may live 5.5 years [10]. Today, extensive molecular and genetic tools are available for zebrafish studies (also to model many human diseases) as they share similar genetic, endocrine, and physiological features to higher vertebrates [11]. Indeed, the zebrafish shares 70% of its genome and over 80% of disease-related proteins with humans [12] and has been used for studying the causal mechanisms of many human diseases as well as for finding new preventive and curative treatments [9,12,13,14]. More than 10,000 researchers [15], belonging to 1595 labs and from 31 countries around the world [16], use zebrafish as animal models. It is estimated that more than 5 million zebrafish are housed in these research facilities [3]. Considering the importance of this animal model and the growing public interest in animal welfare, more attention is sought to optimal zebrafish management, including husbandry and feeding [17].

Nutrition plays a crucial role in maintaining and promoting animal health and welfare. In laboratory conditions, traditional animal models such as rodents are fed with a defined reference diet, using standardized feeding protocols based on an individual’s body weight (BW) [3]. Instead, in many zebrafish facilities, fish are still fed diets formulated for other species (mainly tropical aquarium species) or not suitably shaped (e.g., flakes or pellets rather than extruded diets). Despite the widespread use of zebrafish in research, feeding management is still poorly standardized. For example, nutritional requirements have not yet been defined, and comprehensive feeding protocols are not yet standardized [4,7,18]. Hence, there is a need to develop a widely accepted standard diet for zebrafish; this will minimize unintended nutritional effects on experimental outcomes and facilitate the reproducibility of results across laboratories [4,19,20]. Based on the wider knowledge available for freshwater-farmed fish and other cyprinids (e.g., carp and goldfish), some practical suggestions for feeding zebrafish have been proposed [21,22,23,24]. Unfortunately, these suggestions focused mostly on the juvenile and adult stages, leaving the larval stage largely uncovered. The larval phase represents the most critical period for fish development and survival. Supplying nutritionally balanced diets according to proper feeding protocols (e.g., meal frequency, daily feed quantity) is essential to support larvae growth and health. Many toxicological and preclinical studies are conducted on larvae, and when they are used after exogenous nutrition starts (5 days post-fertilization; dpf), diet composition and feeding management may affect aspects of fish physiology and research outcomes. This is even more important when zebrafish models are used in translational research, as many diseases have nutritional confounders [18]. In fact, nutrition can be responsible for undefined variations in the fish’s biological response. Such variations can be misleading, whereby similar studies using different diets and feeding management protocols may report completely different results and interpretations [4].

The feed ingested by each fish per day or, at least, the feed “properly supplied” (avoiding fish under- or over-feeding), feed intake (FI) is clearly a parameter that needs to be considered when rearing zebrafish. In this regard, very few data are available on zebrafish to date, and published papers rarely mention fish FI. Thus, the aim of this paper was to systematically review the data on zebrafish feeding management, considering the effects of biotic and abiotic factors that may affect zebrafish FI. In addition, a standard zebrafish feeding protocol has been proposed.

## 2. Materials and Methods

### Data Sources and Searches

The search was conducted using the medical library in MEDLINE (via PubMed, last date of consultation 24 January 2023) and the following search strategy: (zebrafish) AND ((feed intake) OR (food intake) OR (feed ingestion) OR (food ingestion)). The resultant abstracts were retrieved, and duplicates were removed. The full text of all potentially eligible articles and their Appendix A were obtained and independently assessed by two authors. Studies were included in the review when they reported quantitative data on FI (the difference between feed supplied and uneaten feed) or feed supplied (decided *a priori*). The exclusion criteria were as follows: (a) duplicates; (b) studies that did not relate to the objective of the article; (c) articles written in a language other than English or Italian; (d) reviews and meeting/workshop abstracts; and (e) books. The data were extracted manually by two authors and were summarized in tables. Data collected included zebrafish FI or the feed supplied (expressed as a percentage of the fish body weight or as mg or number of live prey per single fish per day), animal biological features (line/strain, age, sex), and information on feeding regime and diet nutritional composition (protein, lipids, and energy levels).

## 3. Results

### 3.1. Database Search Process

The PRISMA flow chart of the review process, created following the recent recommendations of Page et al. [25], is presented in Figure 1. The search performed on the PubMed database resulted in a total of 435 records. After the correction of duplicates, 274 records remained. A total of 178 records were excluded because they dealt with a topic different from the topic of interest; 1 manuscript was excluded because it was written in Chinese, 23 were excluded because they were reviews, meeting abstracts, or editorial comments, and 4 were excluded because they were books. Five additional studies were selected by checking the references of the identified relevant papers. A total of 73 studies, from 1994 to 2023, were at last identified for the present review [6,26,27,28,29,30,31,32,33,34,35,36,37,38,39,40,41,42,43,44,45,46,47,48,49,50,51,52,53,54,55,56,57,58,59,60,61,62,63,64,65,66,67,68,69,70,71,72,73,74,75,76,77,78,79,80,81,82,83,84,85,86,87,88,89,90,91,92,93,94,95,96,97,98].

### 3.2. Infographics on Zebrafish and Feeds Data Extrapolated in the Review

Extrapolated data from the reviewed articles were used to generate infographics on zebrafish biological features (line/strain and sex of zebrafish used in these studies), on adopted feeding regimes, and on the nutritional composition of the used diets (e.g., crude protein, crude lipid, and gross energy levels). Regarding zebrafish genotypes (Figure 2A), the zebrafish line/strain was not indicated in about half of the reviewed studies (35 out of 73 articles). The remaining studies focused mainly on the AB strain of the wild-type line. In sexually mature zebrafish, another crucial variable for the interpretation of data is the sex of the studied animals (Figure 2B). After excluding the studies carried out on zebrafish larvae, whose sex cannot be identified, only about one-third of the reviewed studies (25 out of 73) used a balanced male-to-female ratio for evaluating fish FI. In 10 manuscripts, the researchers employed only males since they have fewer physiological requirements (e.g., energy requirements for gonadal development/egg production) compared to females or because they are less resistant to the effects of pollutant exposure. Indeed, female fish usually contain a higher percentage of fat (especially in internal organs), which may dilute lipophilic pollutants [68]. On the contrary, in 8 studies, researchers decided to use only females to minimize male-to-male aggression during the experimental period. The most common feeding frequency regime among the articles reviewed was to feed the animals twice a day (Figure 2C). However, in numerous studies, details were not even reported. Similarly, only 15 studies provided sufficient information on the nutritional composition of the diet, while in most cases, these data were partially available (e.g., only crude protein or crude lipid levels were reported) (Figure 2D). A high variability was found in the nutritional composition of the diets, especially regarding the protein and lipid content (Figure 2E). On average, the crude protein level (% as fed) in the reviewed zebrafish diets was 44.98 ± 9.87 (mean ± SD), while the crude lipid level was 9.91 ± 5.40. Interestingly, the gross energy level of the zebrafish diets was very similar across the reviewed studies, and on average, diets with 20.39 ± 2.10 kilojoules of energy per gram of feed were employed.

### 3.3. Feed Intake Data

Quantitative data on zebrafish FI in the reviewed articles were assigned to two categories: (1) **predefined** when the quantity of feed supplied was decided *a priori* by the researchers, and (2) **calculated** when this value was determined based on the feed voluntarily ingested by fish. In most of the articles reviewed, the zebrafish feed supply was predetermined. In fact, only in 17 articles did the authors record the feed that the fish voluntarily consumed (when provided *ad libitum*). The information retrieved in the review process was organized (Appendix A) and presented graphically in Figure 3. For each manuscript, feeding data (expressed as percentage of fish BW per day) of both categories and fish age were extrapolated and graphed.

The data presented in Figure 3 were then re-graphed by adding a logarithmic trend line to the data points to measure the “goodness of fit” and to better visualize the FI variation during the different growth phases (see Figure 4). The coefficient of determination (R-squared, R^2^) was also calculated for each dataset.

### 3.4. Factors Affecting Zebrafish Feed Intake

The literature review clearly shows that many biotic and abiotic factors can affect zebrafish FI. Drugs and toxicants administration, as well as rearing conditions and genetic manipulations, can affect the dietary intake of zebrafish. In addition, nutrients and nutraceuticals exert a significant effect on feed ingestion. Table 1 summarizes the reviewed information on factors that modulate FI in zebrafish.

## 4. Discussion

Weight gain is often considered the gold standard for the success or failure of a fish diet [4], but it is only one of multiple outcomes that should be considered. Other important performance indicators include survival rate, the incidence of malformations, nutrient utilization efficiency, reproductive performance, gut health, stress resistance, etc. Currently, several companies are producing feeds for zebrafish, promoting their use worldwide. Unfortunately, their composition differs greatly in terms of ingredients, both quantitatively and qualitatively, and this results in differences in fish responsiveness [5,18,19,98].

### 4.1. Zebrafish Feed Categories: What Do They Eat?

The zebrafish is an omnivorous species whose natural diet primarily consists of small aquatic invertebrates [9,11,99]. In laboratory conditions, zebrafish diets can essentially be classified as follows: live prey and dry processed feeds. Live prey (Figure 5) include ciliated protozoan paramecia (*Paramecium* spp.) and *Tetrahymena* spp., rotifers (*Branchionus* spp.), brine shrimp (*Artemia franciscana*, *A. salina*), insect larvae (e.g., *Chironomus* spp.) and micro worms (e.g., *Anguillula aceti*) [7,22,86,100,101]. Also, copepods have been suggested for early-stage freshwater fish due to their unique unsaturated fatty acid profile, high astaxanthin content, triglycerides and phospholipids balance, and high digestibility [102]. Copepods may also be used for adult zebrafish.

Despite their nutritional and ethological benefits (e.g., high digestibility and palatability, the opportunity for nutrient bio-encapsulation, predatory behavior enhancement), live preys bring potential risks for pathogen transmission [86,111], and their use requires significant investments in labor. Moreover, live prey as the sole dietary ingredient can never fully meet fish nutritional requirements as properly formulated dry diets can.

Zebrafish larvae start feeding at 5 dpf. At this stage, small live prey such as *Paramecium* or rotifers is sometimes used in laboratory conditions until 9–15 dpf. When *Paramecium* is used as the sole dietary ingredient, smaller larvae and worse survival rates have been reported [112], and this suggests that *Paramecium* cannot be used as the sole dietary ingredient. *Artemia* nauplii remains the most convenient start-feeding supplement for cyprinid larvae [113,114]. From 15 dpf onwards, zebrafish diets are commonly based on brine shrimps nauplii and dry feeds [5,22]. In general, live preys are commonly used for the first 3 weeks of life, at least [66].

However, dry feeds are considered safer than live prey thanks to their production process, which generally involves a decontaminating heat treatment. Moreover, dry feeds normally have a balanced nutrient profile, are easier to store and supply than live prey, and can be produced with a broad range of particle sizes and shapes. Key aspects when developing fish feeds are feed texture, palatability, color, formulation, and buoyancy, which also affect feed acceptability, digestibility, and nutrient leaching [115]. The original zebrafish husbandry book by Westerfield [22] suggested the use of dry flake feeds. These feeds are still used in some zebrafish facilities, but their use is no longer recommended. Flakes are characterized by substantial nutrient leaching (e.g., water-soluble vitamins) and rapid lipid oxidation compared to extruded or pelleted feeds [116]. These aspects have been confirmed by several studies reporting that adult zebrafish-fed flakes produce significantly fewer eggs than those fed on brine shrimps or other commercial feeds [98,117]. In this context, it has been reported that zebrafish can be reared without using any live prey if the diet is supplied *ad libitum* [19,66,118,119]. Nonetheless, the best results in terms of fish growth and fecundity have been reported when dry feeds are supplemented with live prey, such as brine shrimps [5,66,115,120]. For adult zebrafish only, some authors state that it is possible to attain acceptable levels of performance using only dry feed [5,118].

### 4.2. Zebrafish Feeding Methods and Behaviour: How Do They Eat?

The mechanism regulating fish FI involves several central and peripheral endocrine factors that are affected by variables such as energy reserves, metabolic energy allocation, fish growth, life stage, reproductive status, and environmental conditions. In vertebrates, FI regulating mechanisms are well conserved, notably in fish and mammals [121]. In fish, it has been reported that when unbalanced or pure macronutrient-based diets are offered, individuals can self-select dietary components to properly cover their specific nutritional requirements. This is particularly true in relation to fish dietary protein and energy requirements [122,123,124]. Energy requirements drive FI more than any other nutritional requirements, and when fish are fed nutritionally balanced diets, they regulate FI primarily by meeting their energy needs [125]. Based on this information, an adequate dietary protein-to-energy ratio is a key factor for properly meeting fish protein requirements [126] and avoiding protein excess or deficiency. Similarly, the intake of dietary essential amino acids, essential fatty acids, vitamins, and minerals is also closely related to the dietary energy content [127,128,129]. However, few authors suggest that fish can also regulate FI to meet protein requirements to some extent [41,124,130,131]. Fernandes et al. [41] suggested that FI increases linearly when the level of dietary protein decreases below the fish’s assumed protein requirement. Specifically, when zebrafish were fed diets with a crude protein content lower than 30%, FI increased; on the contrary, when the dietary protein content was higher than 30%, FI remained unchanged. As suggested by Fernandes et al. [41], these results appear to indicate that zebrafish may secondarily regulate FI to meet protein needs. However, another possible explanation may be linked to the composition of the experimental diets and their gross energy content. In fact, to vary the protein content of the diet from 15% to 60%, while maintaining the dietary gross energy level constant (approximately 18.5 kj/g), pre-gelatinized corn starch was used to replace fishmeal. Hence, due to a lower digestibility of this ingredient in comparison to fishmeal, the tested diets were not exactly isoenergetic if their digestible energy content (rather than gross energy content) is considered. If this is the case, the variation in FI can be explained in relation to dietary energy content.

In zebrafish facilities, feeds are supplied using several methods. First of all, it should be considered that zebrafish lack a true stomach, and this must be taken into account when defining feeding frequency [132,133,134,135,136]. Secondly, zebrafish can be fed *ad libitum* or through a predefined ration. *Ad libitum* feeding means that the diet is abundantly available (some feed is always left uneaten), while predefined feeding involves supplying a determined quantity of feed (e.g., 5% of BW per day). When too much feed is supplied, the uneaten feed can adversely affect water quality and fish health by increasing ammonia and reducing water oxygen content [137]. If the feed quantity provided is not adequate, predefined feeding can reduce fish performance (e.g., growth and reproduction). This may happen when the feed that is being supplied is calculated according to the fish BW (e.g., 3 or 5% of the fish BW). Between these two different approaches, a third approach, defined as the “x minutes rule”, is widely used among researchers. This method consists of supplying feed until it is voluntarily consumed with the aim of reaching fish satiety within 5 min. This method was first described by Westerfield [22] as follows: “… *feed manually ground trout pellets, as well as dry flake food, so that all the food is eaten within 5 min, at least twice a day*”. As also reported by Lawrence et al. [138], growth performance is affected by feeding frequency, and this parameter also needs to be considered.

In larval fish culture, using optimal feed particle size increases feeding success and affects growth and survival. Önal [115] showed that at 5 dpf, zebrafish larvae’ mouth gape is 180–200 μm, while at 15 dpf, it is 290–320 μm, and the fish are already able to feed on newly hatched *Artemia* nauplii. Also, Önal [115] reported that 5 dpf zebrafish larvae showed a preference for 21–45 μm feed particle size and were not able to ingest 107–212 μm particles, while at 15 dpf they preferred particles larger than 46–75 μm.

### 4.3. Zebrafish Feed Intake: How Much Do They Eat?

FI is a key aspect of zebrafish nutrition, and it varies according to the fish’s growth and physiological stage. To ascertain whether specific nutritional requirements are met, determining FI would be essential even if a standardized zebrafish diet was used [137]. This is especially true when using an automated feed distribution system, as it means setting a predefined amount of feed to be delivered. Such automated feed distribution systems enable researchers to properly apply feeding protocols while reducing feeding variability and manpower [7]. As previously discussed, FI is affected by levels of stored energy reserves and feed palatability, but it is also affected by biotic factors such as developmental stage, genetics, and health state, as well as environmental abiotic factors, such as temperature and photoperiod, or stress-inducing circumstances [11]. Among environmental factors, temperature plays the main role. As reported by Lu et al. [79], poikilothermic species, such as fish, reduce or even cease feeding when exposed to cold temperatures (see *rearing conditions*—Table 1). The optimal water temperature range for zebrafish is below 31 °C and above 25 °C, with an optimal of 28.5 °C, as also suggested by Westerfield [22].

Accurately measuring FI in zebrafish studies is critical. Previously described methods have included video imaging (e.g., by feeding fish flakes or Paramecia to which fluorescent dyes have been added to measure internalized fluorescence [50,69,73,80,85] or by using photometric assays to measure the orange-red color of fish after feeding brine shrimp [59]) or feed particle or *Artemia* nauplii consumption counting [56,62,70,72,74]. Other authors only considered the time fish spent feeding [53,65]. However, these later methods do not allow for the specific quantification of FI since they do not consider the quantity of feed left uneaten by the fish. Accurate methods for measuring zebrafish FI should always be used in future studies [18] and adequately described in the related papers. This is particularly relevant in the case of preclinical research when genetic manipulation to model human diseases might limit *per se* body size and weight, or behavior, at the larval stage.

### 4.4. Towards a Standard Feeding Strategy for Zebrafish Facilities

The standardization of zebrafish diets and feeding management is critical if the zebrafish is to be effectively used as a model organism [4,17,19]. The implementation of standardized zebrafish management protocols may improve the reliability and effectiveness of studies in various areas, as well as allow for easier and more effective comparisons of the results provided by different authors. Furthermore, the adoption of standardized methods might help minimize both experimental variability and the number of individuals used, the latter in line with the 3Rs principles [139]. Moreover, zebrafish are increasingly being used as a model to evaluate interactions between dietary components (macro- and micro-nutrients) and organism health using genetic and molecular approaches. Thus, the zebrafish is emerging as a new animal model in nutrigenomics and nutrition research, where it has the potential to reveal how dietary treatments affect genes, protein expression, and many physiological traits [7,140,141].

It is commonly accepted that in zebrafish, exogenous nutrition starts at approximately 5 dpf. At this stage, the yolk sac is almost completely consumed, and the development of the digestive system has been completed [142]. In the absence of feed, larvae die within 10–15 days of starvation [5,140]. To date, studies on the FI of zebrafish larvae are not available, and among the reviewed articles, only in two cases were zebrafish larvae considered. In a recent study, 6-day-old larvae were fed 0.3 mg of dry feed per individual per day [80], corresponding to a daily feeding rate of 300% of BW, approximately (at 6 dpf larval BW was estimated to be less than 0.1 mg) [143]. In intensive aquaculture, fish larvae are fed at a rate ranging between 50% and 300% of the BW per day, while adults are fed at a much lower rate, ranging between 1 and 10% [113]. For 25 dpf larvae (BW 5 mg), other authors have reported that each larva consumes 25.3 ± 23.11 (mean ± SD) *Artemia* metanauplii per single meal [70]. However, since no *Artemia* BW was provided, the larvae’s FI remains unidentified. Nowadays, the aquaculture industry is trying to find alternative dry diets to replace live prey such as rotifers and *Artemia* [144,145,146]. The larvae of some freshwater species, such as whitefish (*Coregonus lavaretus*), common carp (*Cyprinus carpio*), ayu (*Plecoglossus altivelus*), and smallmouth bass (*Micropterus dolomievi*), are already commonly fed on dry diets from the start feeding [115]. The aim here is to simplify larvae feeding management, reduce manpower and sanitary risks, and closely meet larvae’ nutritional requirements. From a laboratory management and research perspective, using live prey results in additional difficulties in estimating fish FI.

Based on the references consulted for juvenile and adult zebrafish, FI is higher when it was calculated rather than when fish were fed a predefined quantity of diet (Figure 3 and Figure 4). At the early stages (30–90 dpf), the calculated daily FI ranged from approximately 9% to 5.5% of BW. In contrast, when a predefined quantity of feed was supplied, the feeding rate was 6%. Similarly, at sexual maturation (90–120 dpf), the calculated and predefined feeding rates were both approximately 4.5%. Finally, for adults from 5 to 10 months old, the calculated FI was equal to 4.5%, while the predefined feeding rate varied from 4% to 2.5%. Finally, in the only reference we could find on the use of zebrafish older than 10 months (12 months, precisely), it was reported that the daily feeding rate used was 5% of BW. Otherwise, data from the predefined feed supply analysis were numerically higher, but the reported FI was slightly higher than 2%.

Another important aspect to be considered when assessing zebrafish FI is related to sex. At a similar developmental stage, females are heavier and have higher energy requirements than males, and this results in greater dietary intake [94]. Before carrying out their experiment, Fronte et al. [147] calculated the fish’s FI according to their sex (males and females were separated). This preliminary study was performed to appropriately determine the feed level of a specific ingredient (1,3-1,6 β-glucans) to be supplied according to the fish’s BW. The results of this preliminary study showed that male (BW 391 mg) FI was 3.7% and female (BW 443 mg) FI was 3.9% per day [Fronte, unpublished data]. However, Navarro-Barrón [49] suggested that for adult fish, a maintenance FI of ~2% of BW can meet their caloric requirements without increasing BW.

## 5. Conclusions

Combining the data on zebrafish reference growth curves [133,148,149,150] and the data on calculated FI, a standard feeding strategy has been developed (Figure 6). Considering the negative correlation between feed energy concentration (kilojoule/g) and FI (%/BW), the suggested feeding rates are based on the use of a dry diet containing 20 kj/g (as fed basis) and a water-rearing temperature of 28.5 °C. In detail, for larvae, the suggested daily feeding rate is 9–10% of BW per day, minimum; for juveniles, 6–8% of BW; for adults, 5% of BW. Regarding feed particle size, the following scheme can be used: <100 µm for newly hatched larvae (up to 15 dpf), 100–200 µm between 16 and 30 dpf, 200–400 µm between 31 and 60 dpf, and 400–600 µm from 61 dpf onwards. Moreover, since zebrafish are stomachless foraging fish, it is suggested that providing as many meals as possible per day is advantageous. A good compromise could be 4 meals per day when feed is manually supplied and 6–8 when an automatic feed distribution system is available. Furthermore, the combination of two meals based on dry diets and two meals based on *Artemia* (if possible enriched) can boost fish growth and fecundity. However, when a more precise determination of FI is required due to specific research needs, it is suggested to implement the above feeding protocol with the “five-minute rule” [22]. When performed by trained operators, this practice allows the daily feed ratio (and FI) and the nutritional requirements of the fish to be more precisely met.

It is worthy to highlight that the suggested protocol is based on limited information available to date. Hence, it may be useful updating the proposed feeding protocol in the near future.

Standardizing the zebrafish feeding protocol represents a great challenge and an opportunity. With a particular focus on larvae, specific feeding studies are needed to fully elucidate the effect of feeding practices on growth, health, and voluntary FI. In this regard, it is also suggested that fish genotype, age, and sex must be considered as they are also key variables. To reduce FI variability in a research context and its effect on the observed results, a balanced sex ratio within each group is highly recommended when it is not convenient or possible to separate the sexes.

To conclude, the adoption of the proposed feeding protocol (Figure 6) might represent the first step toward the standardization of feeding procedures within different zebrafish facilities. Pursuing this goal requires strong intra- and inter-laboratory efforts, and this article was written in this spirit and to support this concept.

## Figures and Tables

**Figure 1 biology-13-00209-f001:**
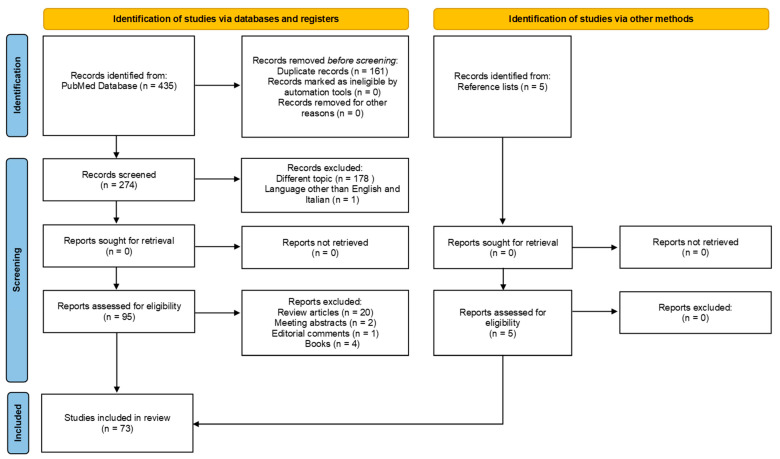
PRISMA flow chart of the review process.

**Figure 2 biology-13-00209-f002:**
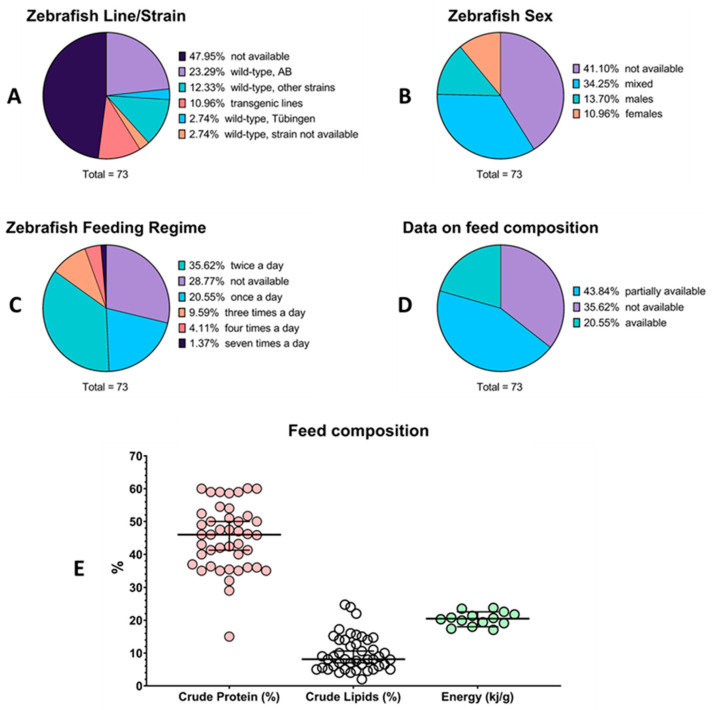
Information gathered from the articles included in the review. (**A**) zebrafish genotype; (**B**) zebrafish sex; (**C**) adopted feeding regime; (**D**) data availability on feed composition; (**E**) quantitative data on feed composition (crude protein, crude lipids, and gross energy) presented as individual values with lines representing mean ± SD.

**Figure 3 biology-13-00209-f003:**
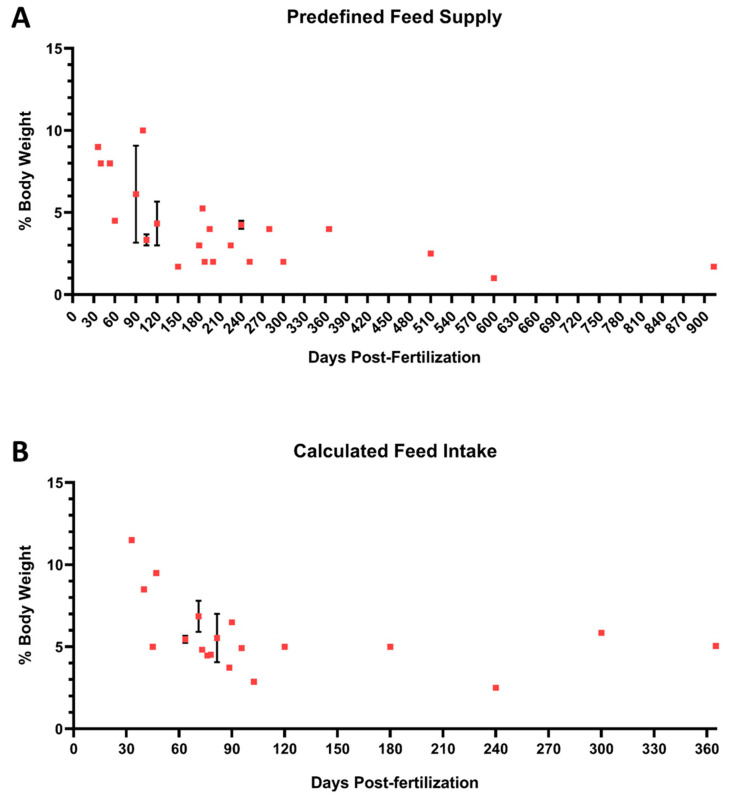
Quantitative data on feed intake from the reviewed articles. (**A**) Data on predefined feed supply and (**B**) on calculated feed intake. Individual values are plotted as means ± SD.

**Figure 4 biology-13-00209-f004:**
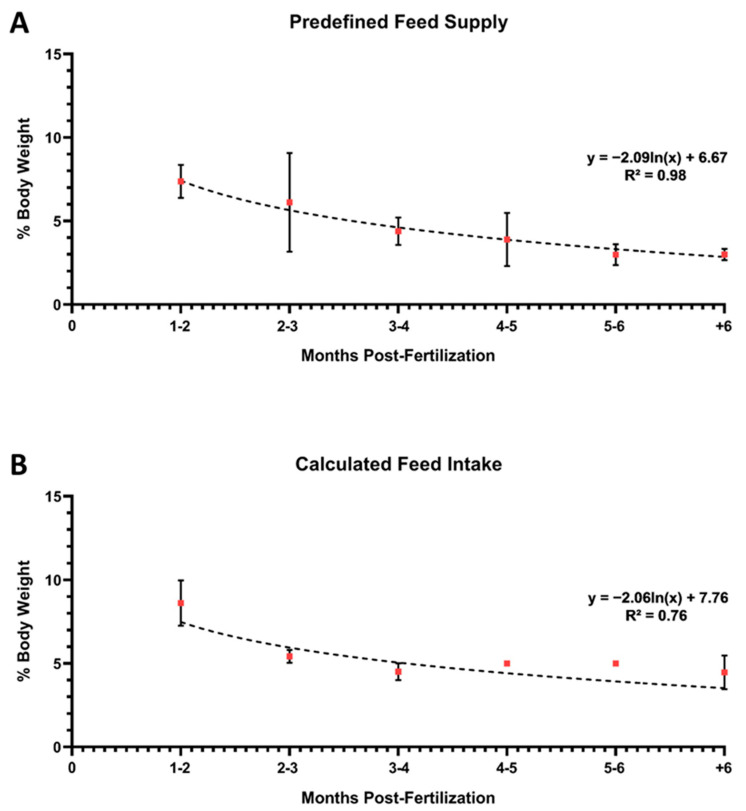
Quantitative data on zebrafish feed intake was retrieved from the articles included in the review. (**A**) Data on predefined feed supply, and (**B**) on calculated feed intake. Values are expressed as means ± SD.

**Figure 5 biology-13-00209-f005:**
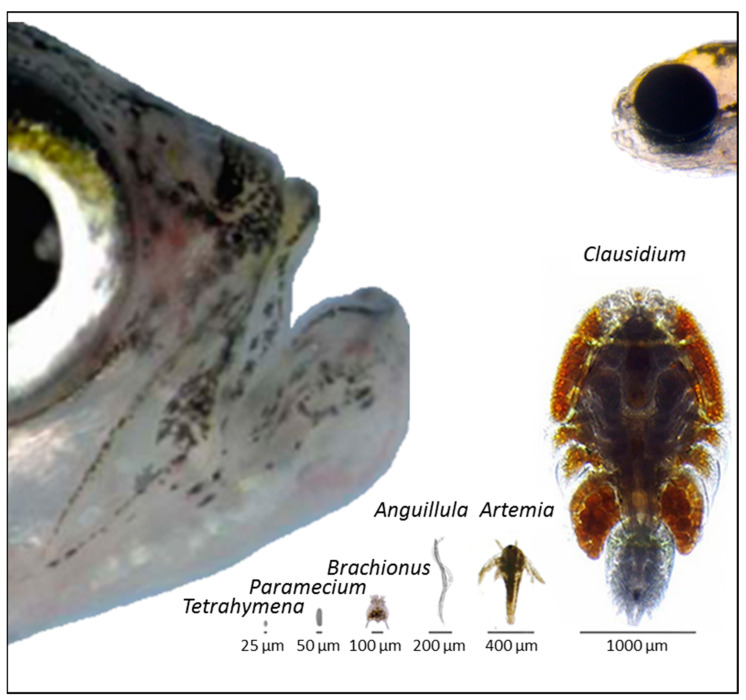
Representative images of live prey conventionally used in zebrafish nutrition: *Tetrahymena* spp. [103], *Paramecium* spp. [104], *Brachionus* spp. [105], *Anguillula* spp. [106], *Artemia* spp. [107], and *Clausidium* spp. [108]. The size of the heads of a 5-day-old larva [109] and an adult fish [110] are shown for reference.

**Figure 6 biology-13-00209-f006:**
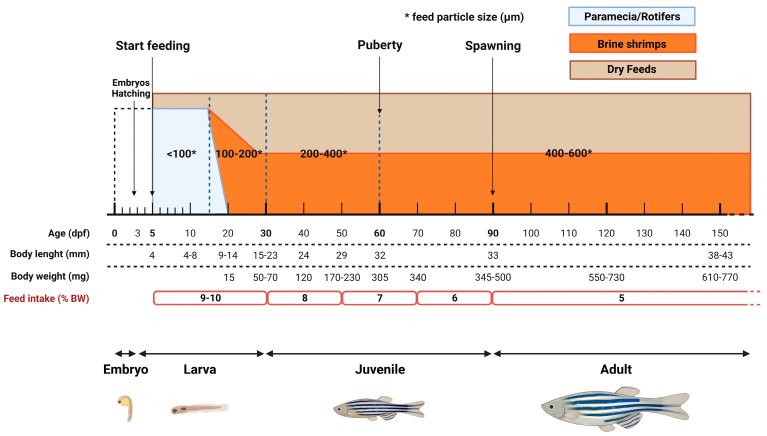
Zebrafish (*Danio rerio*) feeding protocol proposed for standardizing feeding practices. The protocol has been developed based on a feed gross energy content of 20 kilojoule/g (as fed basis) and a water temperature of 28.5 °C. Higher or lower feed energy concentration and water temperature values, may vary fish FI.

**Table 1 biology-13-00209-t001:** Summary of studies describing biotic and abiotic factors affecting zebrafish feed intake.

Reference	Zebrafish Line/Strain	Studied Factor	Effect on Feed Intake
** *drug/toxicant administrations* **			
[27]	n.a.	phoenixin (neuropeptide regulating reproduction, heart, feeding, memory, and anxiety) administration	Reduced with a single intraperitoneal injection of 1 µg/g body weight
[43]	wild-type, AB strain	melatonin exposure	Reduced with melatonin water exposure equal to 100 nM and 1 μM
[45]	n.a.	melatonin exposure	Reduced with melatonin water exposure equal to 100 nM and 1 μM
[48]	wild-type, EK strain	parental whole life cycle dietary methylmercury supplementation	Increased in zebrafish offspring born from parents fed with methylmercury (1, 3 and 10 ppm)
[50]	wild-type	ketoconazole (a fungicidal agent) and tricaine exposure	Reduced with ketoconazole water exposure equal to 10 μM and totally suppressed with tricaine water exposure equal to 380 μM
[53]	n.a.	dietary caulerpin (a bisindole alkaloid extracted from the macroalga *Caulerpa cylindracea*) supplementation	Increased in zebrafish fed caulerpin (0.1%)
[62]	n.a.	pituitary adenylate cyclase-activating polypeptide 1 and 2 (PACAP1 and 2) (neuropeptides activating cAMP production in pituitary cells) administration	Reduced with a single intracerebroventricular injection of zebrafishPACAP1 (2 pmol/g body weight), zebrafish PACAP2 (2 or 20 pmol/g body weight), or mammalian PACAP (2 or 20 pmol/g)
[63]	n.a.	gonadotropin-releasing hormone 2 (decaneuropeptide regulating reproduction and energy balance) administration	Reduced with a single intracerebroventricular injection of 1 pmol/g body weight
[65]	n.a.	carbamazepine (an anticonvulsant) exposure	Reduced with water exposure equal to 10 μg and 10 mg/L
[67]	wild-type	bisphenol A (a plastic additive) exposure	No effect with water exposure equal to 5, 10 and 20 μg/L
[68]	wild-type	bisphenol A and tetrabromobisphenol A (plastic additives) exposure	Increased with water exposures equal to 20, 100, and 500 μg/L
[70]	wild-type	acylated ghrelin (an orexigenic gut hormone) administration	No effect with a single microinjection of 1.1 pmol per egg
[72]	wild-type, AB strain	polystyrene microplastics (alone and coated with bovine serum albumin) exposure	Reduced with water exposure equal to 10 mg/L
[74]	wild-type, AB strain	short and long microplastic fibers exposure	Reduced with water exposure equal to 20 mg/L
[80]	wild-type, AB strain	synthetic phenolic antioxidants (plastics, food packaging materials, petrochemicals, and personal care products additives) exposure	Reduced with water exposure equal to 0.01, 0.1 or 1 μM
[82]	wild-type, AB strain	bisphenol S (a plastic additive) exposure	Increased in female but not in male with water exposures equal to 1, 10 and 100 μg/L
** *rearing conditions* **			
[47]	n.a.	sociality	Reduced when zebrafish are reared alone or in pairs
[64]	wild-type, AB strain	light conditions	Increased using LEDs blue and Reduced using LEDs red, compared to a white fluorescent bulb (control group)
[79]	n.a.	water temperature	Reduced at 22 and 16 °C and stopped at 13 °C compared to controls reared at 28 °C
[83]	wild-type, AB x Tüpfel long fin strain	salt and mechanosensory stress exposures	Reduced with NaCl water exposure equal to 50 and 100 mM and with mechanosensory stress
** *dietary interventions* **			
[6]	wild-type, AB strain	100% replacement of fishmeal with *Hermetia illucens* meal in the diet	No effect with dietary inclusion equal to 5, 10 and 20%
[31]	n.a.	dietary tryptophan supplementation	No effect with dietary inclusion equal to 0.2, 0.6, 1.4 and 3%
[32]	n.a.	100% replacement of fishmeal with soybean meal in the diet	Reduced in zebrafish fed soybean meal-based diet compared to those fed fishmeal-based diet
[34]	wild-type, Tübingen strain	dietary succinate (an acidulant, flavoring additive, and antimicrobial agent) supplementation	Increased in zebrafish fed succinate (0.05, 0.1 and 0.15%)
[35]	wild-type, Tübingen x AB strain	genetically modified feed ingredients inclusion (soya and maize)	Reduced in the groups fed genetically modified soya compared with non-genetically modified soya
[41]	wild-type	dietary protein level	Reduced as the dietary protein level increased up to 35%, remaining stable from this level onward
[97]	wild-type, AB strain	100% replacement of fishmeal with *Hermetia illucens* meal in the diet	No effect with dietary inclusion equal to 17, 33 and 50%
** *genetic manipulations* **			
[28]	wild-type and *gh*-transgenic zebrafish from the F0104 lineage	growth hormone overexpression	Increased in *gh*-transgenic zebrafish line
[30]	wild-type and *gh*-transgenic zebrafish from the F0104 lineage	growth hormone overexpression	Increased in *gh*-transgenic zebrafish line
[46]	*insra*^−/−^ and *insrb*^−/−^ (knockout of insulin receptor a and b), and control (line/strain n.a.)	double knockout of insulin receptor a (*insra*) and b (*insrb*)	Increased in *insra*^−/−^ and *insrb*^−/−^ zebrafish lines
[56]	wild-type AB strain and homozygous *edar* mutants	knockout of ectodysplasin-A receptor (*edar*)	Reduced in homozygous mutants when fed with brine shrimps and Increased in homozygous mutants when fed with dead zebrafish larvae
[57]	*irisin* and control (line/strain n.a.)	irisin (a myokine) administration and knockdown of *irisin* by siRNA	No effect with a single intraperitoneal injection (0.1, 1, 10 and 100 ng/g body weight) Reduced in *irisin* zebrafish line
[69]	homozygous *sgo1* mutant	shugoshin 1 (*sgo1*)	Reduced in homozygous mutants
[73]	heterozygous *smyhc1^mb17/+^*, homozygous *smyhc1^mb1^* and wild-type controls (strain n.a.)	knockdown of slow myosin heavy chain 1 (*smyhc1*)	Reduced in homozygous mutants
[81]	wild-type, Gaighatta—Nadia—Scientific Hatcheries—TM1 strains	4 wild-type strains	Increased in Scientific Hatcheries strain compared to Gaighatta and Nadia strains
[85]	homozygous *mthfr* mutants and control (line/strain n.a.)	knockdown of methylenetetrahydrofolate reductase (*mthfr*) and folic acid exposure	Reduced in homozygous mutants at 5 dpf but not at 8 dpf

Abbreviation: n.a.: not available; dpf: days post-fertilization.

## Data Availability

The original contributions presented in the study are included in the article/Appendix A; further inquiries can be directed to the corresponding authors.

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
