# Peer review of "Zebrafish Feed Intake: A Systematic Review for Standardizing Feeding Management in Laboratory Conditions"

_biology, 2024, doi:10.3390/biology13040209_

Round 1

Reviewer 1 Report

Comments and Suggestions for Authors

The authors address a topic of great importance for zebrafish research, nutrition at different stages of growth. The work done is well presented and easy to read, but it is only a compilation of information without a critical and contrasting analysis of the data compiled. Although a feeding strategy is proposed, it is the one that most people use.  Whether this really covers the nutritional requirements of each developmental stage or it is simply to standardize based on what is most commonly used is not clear.

Author Response

Thank you very much for taking the time to review this manuscript. Please find the detailed responses below and the corresponding revisions in track changes in the re-submitted files.

Point-by-point response to Comments and Suggestions for Authors

Comments 1: The authors address a topic of great importance for zebrafish research, nutrition at different stages of growth. The work done is well presented and easy to read, but it is only a compilation of information without a critical and contrasting analysis of the data compiled. Although a feeding strategy is proposed, it is the one that most people use. Whether this really covers the nutritional requirements of each developmental stage or it is simply to standardize based on what is most commonly used is not clear.

·        Response 1: Thank you for your appreciation and constructive comments on the work. The manuscript did not aim to evaluate the nutritional needs of the species (although the available data were discussed) but to synthesize existing knowledge related to the feeding practices employed in zebrafish facilities, in practical and quantitative terms. Additional goal was to identify areas for further research on zebrafish nutrition. The review serves as a resource for researchers seeking to optimize feeding management strategies and enhance experimental outcomes in zebrafish studies. However, we agree with your comment. Therefore, we have expanded the critical analysis of the reviewed data.

Reviewer 2 Report

Comments and Suggestions for Authors

Despite the need for major revisions, it's important to acknowledge the valuable contribution that the manuscript makes to the field of zebrafish nutrition and husbandry. Overall, the manuscript serves as a valuable resource for researchers seeking to optimize feeding management strategies and enhance experimental outcomes in zebrafish studies. By addressing these points and implementing suggested improvements, the manuscript can be strengthened and better positioned for publication, contributing significantly to the advancement of knowledge in the field of zebrafish nutrition and husbandry.

1.      While the abstract mentions a systematic review of zebrafish feed intake data, it lacks specific details about the methodology employed in the review process. Providing information on the search criteria, inclusion/exclusion criteria, and data analysis methods would enhance the credibility and transparency of the study.

2.      Instead of just stating the number of studies considered (73), it would be more informative to provide summary statistics or key findings from the reviewed studies. For example, highlighting the range of variability in nutrient levels across the diets used and the frequency of biotic and abiotic factors affecting feed intake would add depth to the abstract.

3.      The abstract mentions the proposal of a standard feeding strategy for zebrafish based on the reviewed data, but it lacks details on the specific recommendations or guidelines derived from the analysis. Including key insights or recommendations for researchers would make the abstract more actionable and valuable.

4.      In the introduction text: The transition from discussing the importance of zebrafish in research to the specific topic of nutritional requirements and feeding protocols could be smoother. Consider incorporating a clearer segue to introduce the main focus of the study.

5.      The introduction effectively justifies the need for standardized feeding protocols for zebrafish, highlighting the potential impact of nutrition on research outcomes. However, it could further precisely emphasize the gaps in existing literature and the specific objectives of the systematic review.

6.      Overall, the discussion contributes significantly to the manuscript's objectives by providing a comprehensive analysis of zebrafish feeding practices and their implications for standardized feeding management in laboratory settings. It effectively synthesizes existing knowledge and identifies areas for further research and improvement in zebrafish nutrition and husbandry.

Regards,

The reviewer.

Author Response

Thank you very much for taking the time to review this manuscript. Please find the detailed responses below and the corresponding revisions in track changes in the re-submitted files (new added sentences highlighted in yellow).

Point-by-point response to Comments and Suggestions for Authors

Despite the need for major revisions, it's important to acknowledge the valuable contribution that the manuscript makes to the field of zebrafish nutrition and husbandry. Overall, the manuscript serves as a valuable resource for researchers seeking to optimize feeding management strategies and enhance experimental outcomes in zebrafish studies. By addressing these points and implementing suggested improvements, the manuscript can be strengthened and better positioned for publication, contributing significantly to the advancement of knowledge in the field of zebrafish nutrition and husbandry.

·        Response: Thank you for your appreciation and constructive comments on the work. The main goal of the article was to “photograph” the knowledge currently available on the feeding of the species, in order to critically analyze the most suitable feeding protocols for zebrafish in laboratory conditions.

Comments 1: While the abstract mentions a systematic review of zebrafish feed intake data, it lacks specific details about the methodology employed in the review process. Providing information on the search criteria, inclusion/exclusion criteria, and data analysis methods would enhance the credibility and transparency of the study.

·        Response 1: Specific details about the used methodology were not reported in the abstract, only for reasons of quantitative limitation in the use of words (about 200 words). As indicated in the “Instructions for Authors”, only the main methods are briefly described in the abstract. Detailed informations on the search criteria, inclusion/exclusion criteria, and data analysis were fully described in the paragraph 2.1 “Data Sources and Searches”.

Comments 2: Instead of just stating the number of studies considered (73), it would be more informative to provide summary statistics or key findings from the reviewed studies. For example, highlighting the range of variability in nutrient levels across the diets used and the frequency of biotic and abiotic factors affecting feed intake would add depth to the abstract.

  • Response 2: Thank you for the comment. As suggested, key findings and summary statistics from the reviewed studies were added in the abstract.

Comments 3: The abstract mentions the proposal of a standard feeding strategy for zebrafish based on the reviewed data, but it lacks details on the specific recommendations or guidelines derived from the analysis. Including key insights or recommendations for researchers would make the abstract more actionable and valuable.

  • Response 3: Thank you for the suggestion provided. Practical recommendations were added in the abstract.

Comments 4: In the introduction text: The transition from discussing the importance of zebrafish in research to the specific topic of nutritional requirements and feeding protocols could be smoother. Consider incorporating a clearer segue to introduce the main focus of the study.

  • Response 4: Thank you for this thoughtful comment. A brief introduction to the main topic of the study was added.

Comments 5: The introduction effectively justifies the need for standardized feeding protocols for zebrafish, highlighting the potential of impact of nutrition on research outcomes. However, it could be further precisely emphasize the gaps in existing literature and the specific objectives of the systematic review.

  • Response 5: Thank you for your constructive suggestion. We deeply underline the literature missing points and the practical limitations that complicate nutritional studies on zebrafish.

Comments 6: Overall, the discussion contributes significantly to the manuscript’s objectives by providing a comprehensive analysis of zebrafish feeding practices and their implications for standardized feeding management in laboratory settings. It effectively synthesizes existing knowledge and identifies areas for further research and improvement in zebrafish nutrition and husbandry.

  • Response 6: We thank the Reviewer for her/his positive account of our manuscript and for the in-depth review. Thanks for the kind suggestions provided. We are working with this extraordinary animal every day, by many years. Our goal was to improve the welfare of the species and consequently research outcomes.

Reviewer 3 Report

Comments and Suggestions for Authors 92: 2.1 Data Sources and Searches By limiting the search to the keywords indicated, is it seems possible that relevant papers may have been left out of the study, if they didn't include the exact keywords used in the web search? Was this possibility taken into account and if so, were any measures taken to avoid missing relevant data?   243-244-    "Gonzales [114] also recommended Artemia as the only feed source for rearing juvenile zebrafish."   The former sentence may induce readers to be biased. Other papers suggest that feeding juveniles both dry feeds + live feeds produces better survival and/or growth than using live feeds alone, contradicting Gonzales et al.  Some fish facilities also use rotifers instead of artemia to feed larvae and juveniles up to 60 dpf, while complementing with dry feeds.      244-248 - Some studies indicate that dry feeds alone are better than life feeds alone, regarding survival and growth rate of juveniles.  Authors should include all perspectives and information available in the literature.   Results section:  - Would it be possible to include an analysis of the relation between feed intake % and what feed is provided (eg. live feeds, dry feeds, both)? This is very relevant because, has the authors explained in the discussion, the amount of feed intake varies with the composition of the feeds.  - It would be useful to include a supplementary table with a list of the studies included in each analysis. - In the discussion, lines 414.437, it is included information about larvae feed intake %.  Would it be possible to include this information in the analysis provided in the results, eg. Figure 3 and figure 4?   Section "Recommendations and Conclusions" The protocol proposed on Figure 6 must be subject to review in order to avoid readers from using information wrong. Consider including larger interval of % BW in the proposed feed intake to better represent the available information, and to meet information provided in lines 446-449 and 470-473.   Also in figure 6 the proposed feed intake (% BW) is based on very limited information (eg. lines 446-449). It should be clear to the readers that all the proposed protocols presented on figure 6 are based on limited information and may be adjusted according to the composition of the feeds, physiological status of the animals, etc.      Comments on the Quality of English Language Consider english revision of the following: 72 -  "few" -  consider replacing with "a few" 86-  "aim of this paper is"  consider replacing with "the aims of this paper are"
98-100-  consider rephrasing "Studies were included if they reported quantitative data on food intake (FI) in zebrafish both if this parameter was predefined or if it was methodically calculated"  374-374- consider rephrasing  "measuring the amount of time taken by the fish to feeding"

Author Response

Thank you very much for taking the time to review this manuscript. Please find the detailed responses below and the corresponding revisions in track changes in the re-submitted files (new added sentences highlighted in yellow).

Point-by-point response to Comments and Suggestions for Authors

Comments 1: 92: 2.1 Data Sources and Searches By limiting the search to the keywords indicated, is it seems possible that relevant papers may have been left out of the study, if they didn't include the exact keywords used in the web search? Was this possibility taken into account and if so, were any measures taken to avoid missing relevant data?  

·        Response 1: Thank you for the comment. Of course, we cannot exclude that we might have missed one or more papers, but since we have long experience about this topic, we feel quite confident we did not miss relevant papers. In many other occasions, team members as well as BSc and MSc students, have searched for manuscript related to fish and zebrafish nutrition and so far those mentioned in our paper are those we found. However, if you know we missed any paper, please, let us know and we will be glad to consider it for our review.

Comments 2: 243-244- "Gonzales [114] also recommended Artemia as the only feed source for rearing juvenile zebrafish."   The former sentence may induce readers to be biased. Other papers suggest that feeding juveniles both dry feeds + live feeds produces better survival and/or growth than using live feeds alone, contradicting Gonzales et al.  Some fish facilities also use rotifers instead of artemia to feed larvae and juveniles up to 60 dpf, while complementing with dry feeds.

  • Response 2: Thank you for the comment. First of all, we have modified the sentence, hoping now it is clearer. Regarding the feeding strategies used by the zebrafish community (facilities), as well as the suggestions given by several authors, we totally agreed and are aware of the different approaches and authors’ opinion. However, to this regard we did not suggest to use artemia as solely diets as Gonzales et al. suggested, and that’s why further we discussed about the use of dry diets.

Comments 3: 244-248 - Some studies indicate that dry feeds alone are better than life feeds alone, regarding survival and growth rate of juveniles.  Authors should include all perspectives and information available in the literature.

  • Response 3: Thank you for the suggestion provided. Actually, these aspects has been reported between lines 264 and 268, where it is stated that many authors suggest that the use of dry feed as solely dietary ingredient may be convenient. These authors were cited at line 266. Then, according to your previous comment, we conclude that the best fish performances are achieved when dry feed are supplied together with live preys and authors cited at line 268. However, in consideration of the two comments above, we can argue that even though combining live preys and dry diets frequently lead to better fish performances, this approach is conceptually not totally correct. Dry diets (feeds) are formulated to be complete diet, meaning that they should provide all the nutrients necessary to meet fish requirements and to promote the best fish performances. Unfortunately, this is rarely the case since zebrafish nutritional requirements have not yet been properly estimated and consequently feeds are not correctly formulated. This aspect may explain why some authors suggest using dry diet as solely nutrient sources and some others suggest combining dry diet and live preys. In practice, achieving the best performances depend on the dry diet used.

Comments 4: Results section: Would it be possible to include an analysis of the relation between feed intake % and what feed is provided (eg. live feeds, dry feeds, both)? This is very relevant because, has the authors explained in the discussion, the amount of feed intake varies with the composition of the feeds.  - It would be useful to include a supplementary table with a list of the studies included in each analysis.

  • Response 4: Thank you for this thoughtful comment. This is a good point and we care about it a lot. Unfortunately, usually facilities managers and authors do not provide specific indication about the amount of live preys supplied and indication are given just about dry diets. For this reason, in our review we have considered only studies were dry diets were used. However, according to your suggestion, we have revised the supplementary Table S1, by adding the type of diet.

Comments 5: Discussion - In the discussion, lines 414.437, it is included information about larvae feed intake %.  Would it be possible to include this information in the analysis provided in the results, eg. Figure 3 and figure 4?

  • Response 5: As we mention within lines 417 and 427, data on larvae feed intake are not available, except for two only studies where larvae were fed 300% of their BW and 25 nauplii per larvae, respectively. The other consulted papers do not provide data about larvae feed intake. Hence including this data on a figure (graph) such as figure 3 or 4 is not worthy or feasible.

Comments 6: Section "Recommendations and Conclusions"

The protocol proposed on Figure 6 must be subject to review in order to avoid readers from using information wrong. Consider including larger interval of % BW in the proposed feed intake to better represent the available information, and to meet information provided in lines 446-449 and 470-473.

  • Response 6: The range used in the proposed scheme (figure 6) are derived from the consulted references in which data on calculated feed intake were given. Based on that, we used the calculated mean values for each BW category, in the attempt of suggesting more standardized values and reducing feeding protocol variability within facilities and studies. However, figure 3 and 4 provide enough information for those that would like to adjust these values to their specific conditions.

Comments 7: Also in figure 6 the proposed feed intake (% BW) is based on very limited information (eg. lines 446-449). It should be clear to the readers that all the proposed protocols presented on figure 6 are based on limited information and may be adjusted according to the composition of the feeds, physiological status of the animals, etc.

Response 7: In the caption of the figure 6, it is reported that “higher or lower feed energy concentration may result in lower or higher feed intake”. However, we added a sentence to stress the fact that the proposed data are based on limited information and that it may require to be updated once more information will be available.

Comments on the Quality of English Language: all suggested revisions were amended.

Reviewer 4 Report

Comments and Suggestions for Authors

The authors fail to differentiate between two often misused, yet entirely different terms: feed offered and feed intake.  Feed offered is the amount of feed that is dispensed to fish.  Feed intake is the amount of food ingested by the fish.  Feed intake is relatively easy to determine in terrestrial animals by simply weighing the amount of food offered and subtracting the amount that is not consumed.  However, it is very difficult to determine in aquatic organisms as uneaten feed is difficult to collect, dry, and weigh, particularly with small pelletized diets with water soluble nutrients that cause pellets (or crumbles) to break apart in the water.  This is further complicated by the fact that most aquatic species have a constant water flow in and out of the culture tank, which quickly washes food away.  Alternatives for determining feed intake in aquatic organisms include using fluorescent dies and counting live prey, as mentioned by the authors (lines 369-373).  In line 375 the authors admit that many of the publications they are citing do not properly quantify feed intake. This is true, those studies are reporting the amount of feed offered.  Yet, the authors group those studies (25 publications) together with the studies that actually calculated feed intake (11) and report all as feed intake (see table 1).  This lack of differentiation perpetuates the misuse of these two distinct terms and will mislead readers.  

The authors also provide recommendations for feed intake based on age and size (figure 6).  Feed intake is controlled by multiple factors which include energy requirements, feed palatability, water flow, cohort competition, amount of available food, reproductive status, water chemistry, lighting, etc. Fish care takers have little control of feed intake and monitoring intake is extremely complicated thus these recommendations are trivial. 

The results section 3.1 (Database Search Process) is more appropriate for the Methods section.

Author Response

Thank you very much for taking the time to review this manuscript. Please find the detailed responses below and the corresponding revisions in track changes in the re-submitted files (new added sentences highlighted in yellow).

Point-by-point response to Comments and Suggestions for Authors

Comments 1: The authors fail to differentiate between two often misused, yet entirely different terms: feed offered and feed intake.  Feed offered is the amount of feed that is dispensed to fish.  Feed intake is the amount of food ingested by the fish.  Feed intake is relatively easy to determine in terrestrial animals by simply weighing the amount of food offered and subtracting the amount that is not consumed.  However, it is very difficult to determine in aquatic organisms as uneaten feed is difficult to collect, dry, and weigh, particularly with small pelletized diets with water soluble nutrients that cause pellets (or crumbles) to break apart in the water.  This is further complicated by the fact that most aquatic species have a constant water flow in and out of the culture tank, which quickly washes food away.  Alternatives for determining feed intake in aquatic organisms include using fluorescent dies and counting live prey, as mentioned by the authors (lines 369-373).  In line 375 the authors admit that many of the publications they are citing do not properly quantify feed intake. This is true, those studies are reporting the amount of feed offered.  Yet, the authors group those studies (25 publications) together with the studies that actually calculated feed intake (11) and report all as feed intake (see table 1).  This lack of differentiation perpetuates the misuse of these two distinct terms and will mislead readers. 

The authors also provide recommendations for feed intake based on age and size (figure 6).  Feed intake is controlled by multiple factors which include energy requirements, feed palatability, water flow, cohort competition, amount of available food, reproductive status, water chemistry, lighting, etc. Fish care takers have little control of feed intake and monitoring intake is extremely complicated thus these recommendations are trivial.

The results section 3.1 (Database Search Process) is more appropriate for the Methods section.

·        Response 1: Indeed, under a strict scientific point of view, the term feed intake does not correspond to feed offered (we rather prefer to define this parameter as “feed supplied”). We totally agree on that, and we modified the manuscript accordingly.

However, considered that in aquaculture and zebrafish husbandry there is no suitable and feasible method for measuring fish feed intake during routinary husbandry practices, it is commonly accepted do not differentiate these two terms. In fact, most of the available papers report “feed intake” rather than “feed supplied”. For this reason, considering “trivial” the feed offered parameters provided in the feeding protocol proposed looks neither polite nor wise.

Round 2

Reviewer 2 Report

Comments and Suggestions for Authors

Thanks for the improved version of the manuscript which is hereby recommended for acceptance to publish. 

Author Response

Dear Reviewer,

we are glad you appreciated our effort for improving our manuscipt.

Your suggestions were really worthy.

Best regards.